# The base of the *Lystrosaurus* Assemblage Zone, Karoo Basin, predates the end-Permian marine extinction

Robert A. Gastaldo [1✉], Sandra L. Kamo[2], Johann Neveling[3], John W. Geissman[4,5], Cindy V. Looy [6] & Anna M. Martini[7]

The current model for the end-Permian terrestrial ecosystem crisis holds that systematic loss exhibited by an abrupt turnover from the *Daptocephalus* to the *Lystrosaurus* Assemblage Zone (AZ; Karoo Basin, South Africa) is time equivalent with the marine Permian–Triassic boundary (PTB). The marine event began at 251.941 ± 0.037 Ma, with the PTB placed at 251.902 ± 0.024 Ma (2σ). Radio-isotopic dates over this interval in the Karoo Basin were limited to one high resolution ash-fall deposit in the upper *Daptocephalus* AZ (253.48 ± 0.15 (2σ) Ma) with no similar age constraints for the overlying biozone. Here, we present the first U-Pb CA-ID-TIMS zircon age (252.24 ± 0.11 (2σ) Ma) from a pristine ash-fall deposit in the Karoo *Lystrosaurus* AZ. This date confirms that the lower exposures of the *Lystrosaurus* AZ are of latest Permian age and that the purported turnover in the basin preceded the end-Permian marine event by over 300 ka, thus refuting the previously used stratigraphic marker for terrestrial end-Permian extinction.

[1] Department of Geology, Colby College, Waterville, ME 04901, USA. [2] Department of Earth Sciences, Jack Satterly Geochronology Laboratory, University of Toronto, Toronto, Ontario M5S 3B1, Canada. [3] Council for Geosciences, Private Bag x112, Silverton, Pretoria 0001, South Africa. [4] Department of Geosciences, The University of Texas at Dallas, Richardson, TX 75080-3021, USA. [5] Department of Earth and Planetary Sciences, The University of New Mexico, Albuquerque, NM 87131-0001, USA. [6] Department of Integrative Biology, Museum of Paleontology, University and Jepson Herbaria, University of California–Berkeley, 3060 Valley Life Sciences Building #3140, Berkeley, CA 94720-3140, USA. [7] Department of Geology, Amherst College, Amherst, MA 01002, USA. ✉email: robert.gastaldo@colby.edu

The end-Permian extinction event represents the most catastrophic demise of the Phanerozoic biosphere, with an estimated "instantaneous" biodiversity loss exceeding 90% of marine invertebrate species[1,2] and a reportedly coeval turnover of up to 70% of terrestrial vertebrates[3,4 but see 5]. It is a deep-time model for ecosystem response to increasingly warmer climates, and considered a potential scenario comparable to changes now documented in today's Earth Systems[2,6]. As a consequence of increasing global temperature in the latest Permian (Changhsingian), coincident with major mafic volcanism (>$3.0 \times 10^6$ km³) associated with the emplacement of the Siberian Traps[7,8], ocean temperatures climbed[9], ocean circulation slowed[10], and anoxic waters spread over marine shelves[11]. As a result, pulsed extinction of benthic, nektonic, and pelagic taxa extended into the early Triassic[12].

Postulated global temperature increase over a time interval estimated to range from 60 to 120 ka[4,7] is also thought to have affected and reorganized terrestrial ecosystems[13–15]. Evidence for the effects of increasing aridity are purported to be found in the sedimentologic and paleobiologic records of the Karoo Basin[3,16–18], and elsewhere, and it is thought to have been recorded over a short stratigraphic interval. The Karoo model holds that changes in fluvial architecture, from broad and meandering channels to "braided" regimes[19], resulted from the loss of wetland vegetation[16] which, in turn, reduced resource availability for late Permian vertebrates leading to rapid extinction and turnover[3,4]. These reported Karoo patterns have very limited geochronometric context[20,21] and are not without controversy.

An emerging vertebrate-fossil record, first investigated in the early 20th Century, formed the basis for subdividing the relatively monotonous sandstone-and-siltstone middle Permian to middle Triassic stratigraphy of the Karoo Basin[22,23] (Fig. 1). Currently, eight vertebrate-assemblage zones are recognized as ranging from middle Permian (*Eodicynodon*) into the Triassic (*Cynognathus*), wherein two vertebrate-extinctions are recorded[24]. A middle Permian extinction pulse is identified at the top of the *Tapinocephalus*[24,25] AZ and is documented by several U-Pb ages[26]. In contrast, the reported turnover from the *Daptocephalus* (= *Dicynodon*[27]) AZ to the *Lystrosaurus* AZ, long purported to be coeval with the end-Permian marine extinction interval, has never been well-constrained temporally, although an early Triassic detrital zircon age recently was reported for the upper *Daptocephalus* AZ[18]. The postulated correlation with the marine record currently rests on stable carbon-isotope and magnetic polarity data[3,4, but see 21]. An early Changhsingian age, ~1.5 million years older than the crisis, obtained from an ash fall deposit in magnetostratigraphic context is reported from the upper *Daptocephalus* AZ at one classic Karoo locality[20,21] (Fig. 1). However, locating similar volcanogenic deposits in the *Lystrosaurus* AZ has been challenging. A pristine ash fall deposit on a farm in the Free State Province yields a U-Pb CA-ID-TIMS zircon age of 252.24 ± 0.11 (2σ) Ma demonstrating the reported terrestrial turnover in Gondwana occurred several hundred thousand years before the marine crisis, implying the extinction and turnover mechanisms that operated in terrestrial ecosystems differed from those that operated in the oceans.

## Results

**Nooitgedacht, Free State.** Farm Nooitgedacht 68 lies ~35 km to the NNW of a study area centered on the Bethel 763, Heldenmoed 677, Donald 207 (Fairydale) farms, and Tussen Die Riviere reserve, where 86% of the vertebrate data used to construct the end-Permian model were collected[4,28]. A rich vertebrate assemblage spanning the uppermost *Daptocephalus* and lowermost *Lystrosaurus* AZs recovered from the slopes of two koppies (hills),

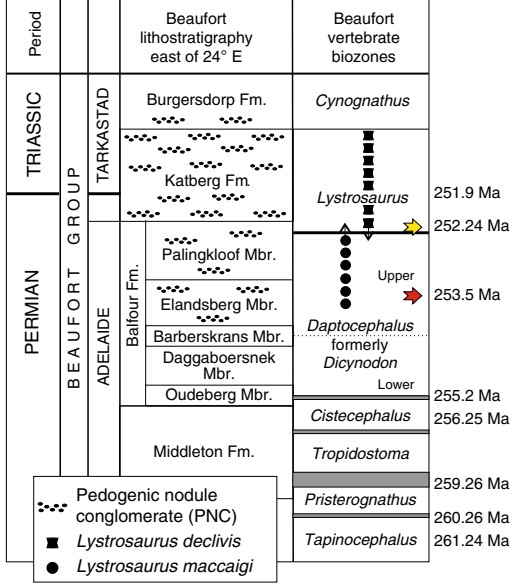

| Period | | | | Beaufort lithostratigraphy east of 24° E | Beaufort vertebrate biozones | |
|---|---|---|---|---|---|---|
| | TRIASSIC | TARKASTAD | | Burgersdorp Fm. | *Cynognathus* | |
| | | | | Katberg Fm. | *Lystrosaurus* | 251.9 Ma |
| | | | | | | 252.24 Ma |
| | PERMIAN | ADELAIDE | Balfour Fm. | Palingkloof Mbr. | | |
| | | | | Elandsberg Mbr. | Upper | 253.5 Ma |
| | | | | Barberskrans Mbr. | *Daptocephalus* formerly *Dicynodon* | |
| | | | | Daggaboersnek Mbr. | | |
| | | | | Oudeberg Mbr. | Lower | 255.2 Ma |
| | | | | | *Cistecephalus* | 256.25 Ma |
| | | | | Middleton Fm. | *Tropidostoma* | |
| | | | | | *Pristerognathus* | 259.26 Ma |
| | | | | | *Tapinocephalus* | 260.26 Ma |
| | | | | | | 261.24 Ma |

• • • Pedogenic nodule conglomerate (PNC)
■ *Lystrosaurus declivis*
● *Lystrosaurus maccaigi*

**Fig. 1 Generalized stratigraphy of the Permian–Triassic Beaufort Group, Karoo Basin, South Africa, with vertebrate biozones.** U-Pb age assignments separating the *Tropidostoma*, *Cistephalus*, and *Daptocephalus* AZs from Day et al.[24]. and Rubidge et al.[26]; red arrow is lower Changhsingian age in the Elandsberg Member[20]; yellow arrow is date reported, herein; and published postulated position of the Permian–Triassic boundary age (as based on the marine record[7]). Vertebrate-range extensions follow re-analysis of original data[28] set used to interpret the terrestrial response to the end-Permian event[3,4].

referred to as Loskop and Spitskop (Supplementary Fig. 1), are inferred to represent a complete Permo–Triassic Boundary (PTB) sequence[29]—but see geologic evidence for extensive unconformities contained in this stratigraphy[20,21,30]—and the locality has been utilized in studies on the extinction dynamics in the basin[29,31]. A rich assemblage of *Lystrosaurus maccaigi*, considered to be diagnostic of the pre-extinction fauna[4,29,32,33], is overlain by two "marker" taxa, *Lystrosaurus curvatus* and *Moschorhinus kitchingii*, that have been used to support the placement of the "PTB" at the top of a so-called heterolithic facies[18,29,31]. A lower "PTB" position was recently proposed[18], but the former[29] is used to delineate the vertebrate biozone boundary in the current study (Fig. 2).

Similar to other Changhsingian successions in the Karoo Basin, Nooitgedacht's stratigraphy comprises a seemingly monotonous succession of fine- to very-fine grained feldspathic wacke and mudrock, arranged in upward-fining cycles (Fig. 2a–c; Supplementary Note 1). Mudrock intervals range from coarse to fine siltstone, varying from greenish gray to grayish or brownish red in color[33], which are intercalated with sandstone bodies. Mudrock in *Daptocephalus* AZ strata varies in color from brownish/reddish gray and dusky red to olive and light-olive gray. Upsection, mudrock displays increasingly reddish and brownish colors, while isolated and amalgamated sandstone lenses increase in abundance. These sandstones thin and split laterally, grading into thinly bedded, reddish gray and olive gray lenses of (coarse) siltstone; some of these heterolithic intervals have been interpreted to mark the PTB[18,29] (Fig. 2a–c, red arrow). The overlying succession is inferred to be earliest Triassic based on the prevailing vertebrate biostratigraphic model, and is characterized by dusky red and olive-gray laminated mudrock that grades upwards into a sheet-sandstone succession (Fig. 2c). The latter consists of several fining-up cycles, with fine- to very

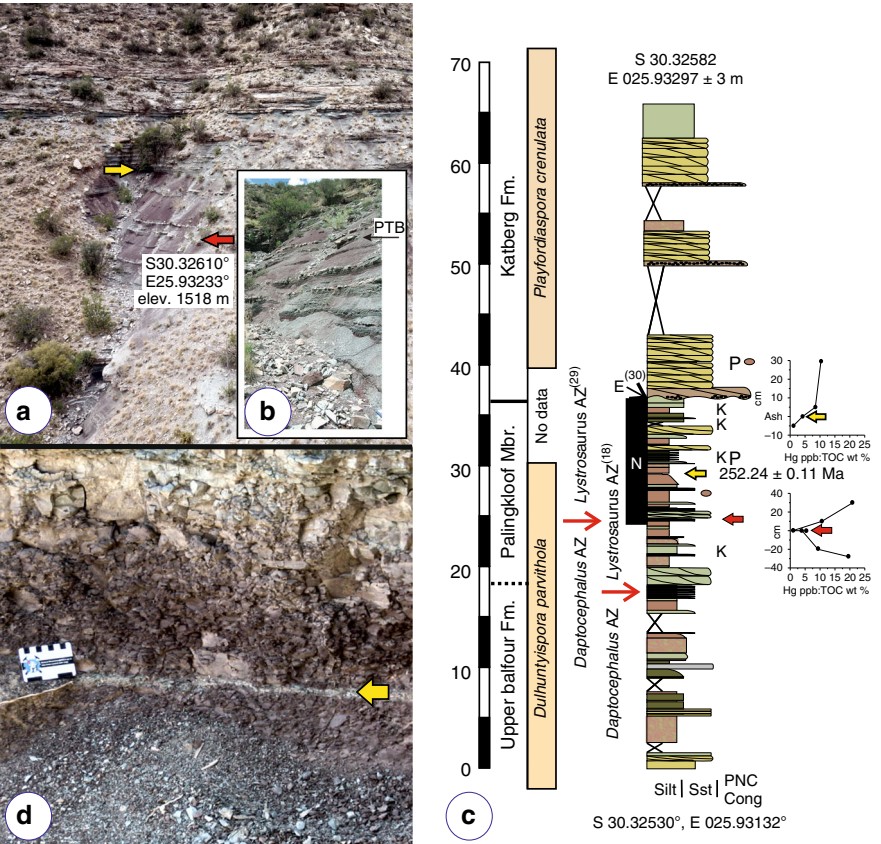

**Fig. 2 Nooitgedacht stratigraphic section. a** Orthogonal outcrop image using drone technology (see Supplementary Note 1) in which the red arrow marks the horizon identified by other workers as the terrestrial PTB and the yellow arrow indicates the location of the ash-fall bed from which our U-Pb CA ID-TIMS age assignment originates. **b** Image from Botha-Brink et al.[29] identifying their vertebrate-defined PTB; compare with (**a**). Note that Botha et al.[18] lowered their vertebrate-defined PTB ~8 m in the section. **c** Measured stratigraphic column beginning at an exposed, resistant sandstone body in the Loskop koppie. Red arrow marks the vertebrate-defined PTB as above with biozone boundaries as reported in 2014[29] and 2020[18] for the same reported stratigraphic section; yellow arrow marks the ash-fall horizon; P marks the palynological assemblages; a normal magnetic polarity zone is marked, accordingly; E identifies an erosional contact marking a phase of landscape degradation and missing section[30]; and plots of Hg ppb:TOC % for two intervals are provided, with vertical scale in centimeters. Hg and TOC values are the average of triplicate analyses of each horizon sampled. Palynological zone assignments, correlated with Australia[39,40], appear against recovered pollen-and-spore assemblages. See supplemental information for legend. Vertical scale in 5 m intervals. **d** Field image of thin, very light gray (N8) ash deposit sampled in the current study. Scale in cm.

fine-grained feldspathic wacke in which intraformational conglomerate lags occur infrequently[21,30]. The first thick sheet sandstone occurs approximately 12 m above the proposed "PTB"[29] in our section (Fig. 2). Within the dusky red and olive-gray laminated mudrock interval and below the first thick channel-fill, we have identified a well-exposed thin, up to ~1.0 cm thick ash bed (Fig. 2a, d yellow arrow; Supplementary Fig. 2), from which we have recovered a population of pristine, euhedral zircon grains.

**U-Pb CA-ID-TIMS results**. Reported U-Pb dates are based on the $^{238}U$–$^{206}Pb$ decay scheme. This is the most robust system for geologically young rocks due to the greater abundance of $^{238}U$ and ingrown radiogenic $^{206}Pb$. Results of fourteen single crystal zircon, CA-ID TIMS analyses are presented in Supplementary Table 1, and a concordia diagram and plot of individual $^{206}Pb/^{238}U$ zircon-crystal dates are presented in Fig. 3.

About 1000 zircon grains were recovered from ~800 g of the ash bed and ~100 and ~220 detrital zircon grains were recovered from the enveloping brownish/reddish gray siltstone above and below the ash bed, respectively. Zircon grains recovered from the ash bed are consistently euhedral, translucent, and typically long

prismatic, 2/1 short prismatic, or equant and multi-faceted (Fig. 3; lower left insert). In contrast, detrital zircon grains show wide variation in color, morphology, and grain size, with variable degrees of surface abrasion and rounding. Only euhedral, prismatic, and multi-faceted grains, in which small melt inclusions may be present, were analyzed. U-Pb data obtained from 13 single, chemically abraded zircon grains show overlapping and concordant results (Fig. 3). This is consistent with petrographic observations showing what we interpret to be a primary, depositional fabric for the ash deposit (Supplementary Fig. 3). These have a weighted mean $^{206}Pb/^{238}U$ age of 252.24 ± 0.11 Ma (2σ, MSWD = 0.58). Data for four analyses were obtained using an in-house (ROM) spike (mixed $^{205}Pb$-$^{235}U$) (shown as light-gray bars Fig. 3), which give three oldest and one youngest dates. If these four dates are excluded from the mean and only dates obtained with the community EARTHTIME (ET535) spike (mixed $^{205}Pb$-$^{233}U$-$^{235}U$) are considered, the age is ~10 ka younger at 252.23 ± 0.12 Ma (2σ, MSWD = 0.54). An older xenocrystic grain, z1, yields a date of 254.2 ± 0.7 Ma and plots just outside of the analytical uncertainty of the data cluster. Th/U ratios vary between 1.4 and 1.7 for all results with the exception of z2 and z14, which are somewhat lower and higher, respectively, at 0.94 and 2.3. We conclude the most robust age

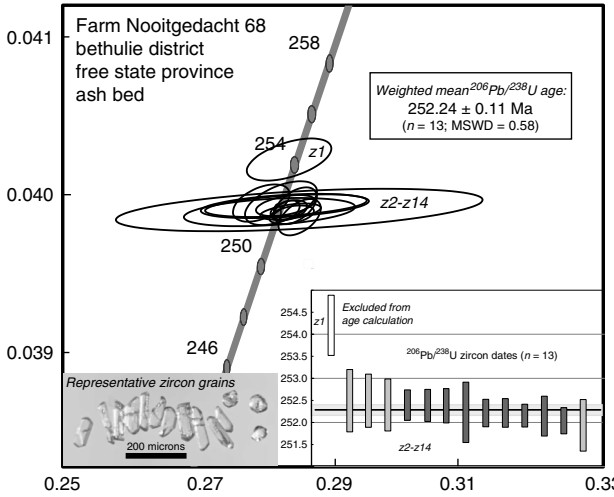

**Fig. 3 A Wetherill concordia diagram showing U-Pb ID-TIMS data for single zircon crystals from the ~1 cm thick ash bed, Nooitgedacht section, Karoo Basin.** A Wetherill concordia diagram plots U-Pb (uranium-lead) data, and associated error ellipses for Pb/U ratios based on 2-sigma errors, with $^{206}Pb$-$^{238}U$ ratios on the Y axis and $^{207}Pb$-$^{235}U$ ratios along the x axis. Our Fig. 3 does not include individual points (centroids to the ellipses) as these would be obscured due to the degree of overlapping of our ellipses. The weighted mean age obtained from data for 13 individual grains is 252.24 ± 0.11 Ma (2 σ; MSWD = 0.58). This excludes the data for one older xenocryst, which was omitted from the mean age. Lower right inset shows plot of $^{206}Pb$/$^{238}U$ dates, with horizontal gray bar indicating two-sigma error range of mean. Overlapping dates obtained with EARTHTIME tracer ($^{205}Pb$-$^{233}U$-$^{235}U$) shown in dark gray bars and those from ROM tracer ($^{205}Pb$-$^{235}U$) in lighter gray bars. Lower left inset is an image of zircon grains representative of the dated grains and the population, in general.

estimate for deposition of the ash-fall layer is 252.24 ± 0.11 Ma, based on all 13 dates.

**Magnetostratigraphy**. To determine the magnetic polarity of the stratigraphy immediately above and below the ash fall deposit, we sampled the same enveloping brownish/reddish gray siltstone within about 5 cm above and below the ash bed (see Supplementary Note 2 for methodology using ceramic cube sampling results from thermal demagnetization and bulk susceptibility experiments). Response to progressive thermal demagnetization by the siltstone is of high quality, with the typical isolation of a component of magnetization over a range of laboratory unblocking temperatures to ~680 ºC (Supplementary Figs. 4, 5; Supplementary Table 2) implying hematite as the key carrier of the remanence in these rocks, as supported by rock magnetic experiments (Supplementary Fig. 6). Samples from siltstone immediately above the ash bed yield magnetizations that are typically of steep negative inclination yet somewhat dispersed declinations. Those samples collected from below the ash bed yield magnetizations of moderate to steep negative inclination with greater dispersion (Supplementary Figs. 4, 5). We interpret these magnetizations to be of normal polarity, and that the observed dispersion in directions is a function of the more unorthodox means by which the siltstones were, by necessity, sampled (see Supplementary Note 2). The demagnetization data give no hint of the preservation of a magnetization of opposite (reverse) polarity in these rocks. Assuming that the magnetizations characteristic of the siltstones are early-acquired (i.e., primary) then the normal polarity magnetozone, with the upper and lower boundaries that remain undetermined, that is

indicated by these data can be correlated with a normal polarity chron in geomagnetic polarity time scales that have been compiled for the interval near the PTB[34,35]. The correlation indicates that either the magnetozone represents the very earliest part of the normal polarity chron in which the PTB is defined to lie in the marine realm or it is part of the previous (older) normal polarity chron (Supplementary Fig. 7). The latter option is, at present, deemed less likely. In comparison to magnetic polarity stratigraphic data from the nearby Bethel farm[28], the normal polarity magnetozone at Nooitgedacht lies above the horizon where the biozone contact is identified to lie within a reverse polarity zone. This interpretation is based on the current definition of the vertebrate biozones.

**Mercury**. A strong signal of elevated Hg/TOC (>200–1000 ppb/wt.%) has been identified at many PTB sites worldwide, and linked to the emplacement of the Siberian LIP[7,8]. All Hg/TOC values from siltstone sampled across the biozone boundary[29] and the ash bed are below 30 ppb/wt% (Fig. 2; Supplementary Note 3 and Supplementary Table 3), with the lowest values obtained directly at the inferred biozone boundary. In contrast, elevated Hg:TOC values are reported in the Global Boundary Stratotype Section and Point (GSSP) at Meishan that are up to 900 ppb/wt.% relative to a background of <100 ppb/wt.%[36]. Hence, no evidence exists for mercury enrichment in the intervals sampled in the Nooitgedacht section.

**Palynology**. Palynomorph assemblages were recovered from two stratigraphic positions above the ash bed (Fig. 2, Supplementary Fig. 2). The palynomorph assemblage immediately above the ash bed, at a stratigraphic height of 29.9 m (Fig. 2), is of low diversity and dominated by algal remains (*Leiosphaeridia*) and simple spores. The sample contains low amounts of pollen (*Protohaploxypinus*) produced by glossopterids (Supplementary Note 4 Information; Supplementary Fig. 8). The sample at a stratigraphic height of 40.42 m is also dominated by algal remains with simple spores (e.g., *Brevitriletes*, *Horriditriletes*) and sulcate pollen (*Cycadopites*) being the dominant terrestrial components (Supplementary Fig. 9). Of interest is the absence of glossopterid pollen and the presence of the gymnospermous pollen *Ephedripites*, *Cycadopites*, *Falcisporites*, *Lueckisporites* and the cavate spore *Densoisporites nejburgii*. The latter palynomorphs represent gymnosperms such as conifers, peltasperms, corystosperms, and cormose lycophytes. Our two assemblages represent the transition of South African basinal lowland communities in which *Glossopteris* were still present to those characterized by other gymnosperms, cormose lycopsids, and ferns. This floral change is observed in many areas of Gondwana and was, until recently, interpreted as the terrestrial equivalent of the marine end-Permian biotic crisis[37,38]. More recent discoveries from the Sydney Basin in Australia, however, showed that the collapse of glossopterid dominated forests occurred about 370k years prior to the marine event[6], although we note the presence there of a long diastem between the last Permian taxa characteristic of the *Glossopteris* flora and the first post-extinction pollen record[39,40].

**Discussion**

For over a century[22], the vertebrate-fossil record of the Karoo Basin has been held as the world's standard for interpreting the effects of the end-Permian crisis on terrestrial vertebrates[41], the landscapes they inhabited[3,4,32], and the ecosystems in which they lived[15]. According to this model, the last vestiges of the Permian fauna are restricted to the *Daptocephalus* (=*Dicynodon*) AZ, whereas the overlying *Lystrosaurus* AZ, with its purported rapid vertebrate recovery and diversification[4,31], has been assigned an

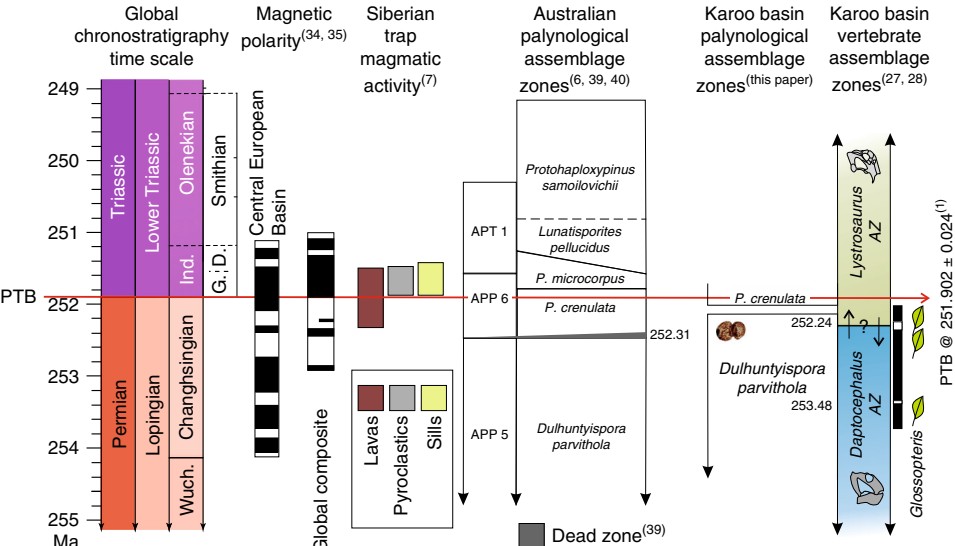

**Fig. 4 Diagram synthesis of late Permian and early Triassic global chronostratigraphic time scale in Ma; magnetostratigraphy and polarity intervals[34,35] (black = normal, white = reverse); duration of Siberian Trap magmatic lava, pyroclastic, and sill emplacement activity[7]; Australian palynological assemblage zones[39,40] and geochronometric age placed on vegetation collapse in the Sydney Basin;[6] compared with the results of the present study.** Ages of the *Dulhuntyispora parvithola* palynological assemblage are provided for the upper *Daptocephalus*[20,21] and lower *Lystrosaurus* AZ[(this paper)] against magnetostratigraphic context based on interpreted magnetic stratigraphy at Old Lootsberg Pass and Bethel farm[21,28]. *Striatopodocarpites fusus*, a taxon of the *D. parvithola* zone, is illustrative of taeniate pollen grains. Biostratigraphically important *Daptocephalus* AZ and *Lystrosaurus* AZ taxa range extensions, up and down, follow an assessment of the original data set used to construct the end-Permian extinction model[28]. *Glossopteris* leaves are preserved as macrofossils in the upper *Daptocephalus*[58-60] and lower *Lystrosaurus* AZs[60]. Global Stages: Wuch. = Wuchiapingian, Ind. = Induan; European Triassic stages: G = Griesbachian, D. = Dienerian.

earliest Triassic age. This paradigm is extrapolated across the southern Gondwanan continents including Antarctica[42], South America[43], Laos[44], and India[45], and into the northern hemisphere including Angara[46] and Cathaysia[47].

The terrestrial calamity has been correlated with increasing global aridity in response to continued Siberian Trap magmatism, and considered to be time equivalent with the latest Permian phased extinctions in the marine record. Our late Changhsingian age from an ash bed in strata of the lower *Lystrosaurus* AZ at Nooitgedacht, some 340 ka older than the main marine extinction pulse[1], indicates that the vertebrate-biozone boundary, irrespective of placement[18,29], in the Karoo Basin, as currently recognized[27,32], is not coincident with the marine crisis (Fig. 4). Hence, interpretations about behavioral[48-50], physiological[50-52], biogeographical[53], extinction mechanisms[54], and life-strategy patterns[55-57] of the *Lystrosaurus* AZ tetrapods do not reflect a response to, or consequence of, the terminal marine crisis. Yet, our high precision age determination from the Nooitgedacht section and corresponding palynological record does conform with recent findings in eastern Australia[6].

A U-Pb CA-ID-TIMS age of 252.31 ± 0.07 Ma is reported and interpreted to constrain the regional collapse of the *Glossopteris* flora in the Sydney Basin[6]. This age estimate, too, is ~370 ka prior to the onset of the marine extinction interval but concurrent with the onset of the initial pulse of Siberian Trap volcanism[7,8]. The Australian date is statistically indistinguishable from our age determination for the ash deposit in the Nooitgedacht section. Based on the proportion of major pollen-and-spore categories and several marker taxa, the Karoo palynoassemblages are considered to be contemporaneous and equivalent to two Australian zones. The pollen assemblage immediately above the ash-fall tuff (Fig. 2) is correlative with the *Dulhuntyispora parvithola* zone[39,40] and the assemblage preserved stratigraphically higher is correlative with the *Playfordiaspora crenulata* zone[39,40]. It is at the base of the *P. crenulata* zone that a major floral change is interpreted

as the demise of the *Glossopteris* flora in the Sydney Basin, Australia (Fig. 4). Assemblages characterized by taeniate bisaccate *Protohaploxypinus* and *Striatopodocarpidites* pollen are replaced by assemblages rich in algal remains and low abundance of non-taeniate, alete bissacate pollen and cavate spores. There is an absence of *Glossopteris* macroflora above that horizon in Australia, although leaves of this taxon are preserved in the *P. crenulata* zone of the Karoo[21,58-60] indicating their persistence in southern Africa. The *P. crenulata* zone is recognized as latest Changhsingian in age, and does not represent the typical Late Permian *Glossopteris* palynofloras. This is consistent with palynoassemblages documented several tens of meters higher in the Katberg Formation exposed on the Donald 207 (Fairydale) farm[28].

The age obtained from the ash deposit at Nooitgedacht necessitates that the *Daptocephalus–Lystrosaurus* faunal boundary be decoupled from the end-Permian extinction in the marine realm. This result also calls into question the prevailing paradigm for late Permian terrestrial ecosystem perturbation. Instead of the currently favored paradigm of calamitous and globally synchronous turnover in ecosystems[3,18,19,41], the reported terrestrial turnover in Gondwana occurred hundreds of thousands of years before the marine one and, therefore, marine and terrestrial responses likely had different extinction mechanisms. Hence, a detailed comparison of terrestrial successions, made possible by high-precision U-Pb geochronology, suggests that greater consideration should be given to a more gradual, complex, and nuanced transition of terrestrial ecosystems during the Changhsingian and, possibly, the early Triassic.

## Methods

**U-Pb CA ID-TIMS Geochronology.** About ~800 g of very fine-grained, light green ash from a ~1-cm-thick ash-fall layer was sampled with care taken to exclude material from adjacent beds and/or loose detritus that could contain zircon grains. The layer is hosted in a ~0.5-m-thick, flat-lying, massive, reddish-gray siltstone

unit on Farm Nooitgedacht 68, Bethulie District (S30.32616°, E025.93242°). For comparison of zircon populations, similar-sized samples of the massive red siltstone were collected from immediately above and below the ash bed.

The samples were disaggregated in a ring mill and a heavy mineral concentrate was produced on a Wilfley table. This was followed by standard mineral-separation procedures using magnetic (Isodynamic Frantz) and heavy liquid (methylene iodide) methods, the latter in small (~10 mL) centrifuge tubes.

U-Pb analysis was by isotope dilution-thermal ionization mass spectrometry methods on single chemically abraded zircon grains (CA-ID-TIMS) in the Jack Satterly Geochronology Laboratory of the Department of Earth Sciences at the University of Toronto. Prior to dissolution and analysis, zircon crystals were thermally annealed at 900 °C for 48 h to repair radiation damage in the crystal lattice. Subsequently, the grains were partially dissolved in ~0.1 ml ~50% hydrofluoric acid and ~0.020 ml of $HNO_3$ at 200 °C for 9 h[61]. Zircon grains were rinsed with 6 N HCl followed by 8 N $HNO_3$ at room temperature prior to dissolution. A $^{205}Pb$-$^{233-235}U$ spike from the EARTHTIME Project or an in-house $^{205}Pb$-$^{235}U$ (ROM) spike was added to the Teflon dissolution capsules during sample loading. Zircon was dissolved using ~0.10 ml of concentrated HF acid and ~0.020 ml of 8 N $HNO_3$ at 200 °C for 5 days, then dried to a precipitate and re-dissolved in ~0.15 ml of 3 N HCl at 200 °C overnight[62]. U and Pb were isolated from the zircon using 50 µl anion exchange columns using HCl, deposited onto outgassed rhenium filaments with silica gel[63], and analyzed with a VG354 mass spectrometer using a single Daly detector in pulse counting mode for Pb, and three Faraday cups in static analysis mode for U or Daly detector if the signal was <300KCps. Corrections to the $^{206}Pb$-$^{238}U$ ages for initial $^{230}Th$ disequilibrium in the zircon have been made assuming a Th/U ratio in the magma of 4.2. All common Pb in each analysis was assigned the isotopic composition of procedural Pb blank. Dead time of the measuring system for Pb was 16 ns. The mass discrimination correction for the Daly detector is constant at 0.05% per atomic mass unit; the thermal mass fractionation correction for Pb was 0.10% per atomic mass unit (±0.076%, 2σ); and the U thermal mass fractionation correction was measured and corrected within each measurement block for static runs. Amplifier gains and Daly characteristics were monitored using the SRM 982 Pb standard. Decay constants are those of Jaffey et al.[64] Age errors quoted in the text and Supplementary Table 2, and error ellipses in the concordia diagram and weighted mean age plot (Fig. 3) are given at the 95% confidence interval. Plotting of U-Pb data employed Isoplot 3.76[65].

**Magnetic properties and petrographic inspection**. We sampled siltstone beds within 5 cm above and below the ~1 cm ash-fall bed sampled for geochronology, within the massive, ~0.5 m thick red siltstone interval. These intervals are highly weathered and fragmentary. After cleaning off the exposure with non-magnetic implements, small (<0.7 cm) chips of thinly bedded siltstone were carefully placed into ceramic boxes (measuring 1.7 cm on a side; Beijing Eusci Technologies Ltd.) with ceramic lids, keeping each chip upright and roughly oriented with respect to Geographic north. Chips were removed using non-magnetic tweezers, and, if needed, they were shaped into appropriate size using non-magnetic (Cu-Be) tools. Glass wool or cotton was used to pack the chips in the ceramic boxes, to prevent movement or fragmentation during transport, and the boxes taped shut. In the laboratory, the glass wool or cotton was removed and the ceramic cubes filled with Zircar alumina cement, which is completely non-magnetic. The ceramic boxes were labeled using a soft aluminum rod, and then subjected to progressive thermal demagnetization using an ASC TD48 thermal demagnetization unit. Magnetizations were measured on a pulse-cooled DC SQUID 2G Enterprises magnetometer. Demagnetization data were inspected using orthogonal demagnetization diagrams[66] and directions of components of magnetization were determined using principal components analysis[67]. The general dispersion (in declination) of magnetizations isolated in these materials is largely attributed to the nature of the sampling procedure, necessitated by the very fissile and friable nature of the hematitic siltstone.

**Petrographic preparation**. An intact sample of the ash-fall layer was prepared for petrographic inspection as follows. After collecting sufficient material from the layer for geochronologic analysis, we carved out the siltstone interval immediately above the ash-fall layer to form a bench indented some six to eight cm into the exposure that preserved the entire interval of ash together with immediately underlying siltstone. A dilute epoxy resin solution was poured onto the bench, in several stages over several hours to ensure that the ash and underlying siltstone was encased in resin. The hardened sample was removed from the exposure 24 h later. Large-format polished petrographic thin sections were prepared from the sample, to include the complete, intact ash-fall layer, the underlying, intact siltstone interval, and fragments of the overlying siltstone interval.

**Mercury analyses**. Analyses were completed on whole-rock powders and, for carbonate containing samples, on residual powder de-carbonated with hydrochloric acid. Organic carbon content was measured on a Costech ECS 4010 elemental analyzer (EA). Hg was measured with a Teledyne Leeman Labs Hydra IIc mercury analyzer.

Relative standard deviation was <4.1% for the EA using a pure methionine standard. For the Hydra IIc it was <9% using the NIST 2702 standard.

**Palynological analyses**. Siltstone samples for palynological analysis were prepared by RPS Laboratory, Northwich, Cheshire, United Kingdom, and residues sieved at 15 µm. Slides are curated in the Museum of Paleontology, University of California, Berkeley, California, under the locality Nooitgedacht 68 – UCMP PA1378, with PA1378.01 (40.42 m) and PA1378.02 (29.9 m). Preparations from horizons 30.35 m and 30.42 m yielded very low numbers spores and pollen from which it was not possible to evaluate either assemblage. Slides are N2990 P-1 and N2990 P-2, and N4042 P-1 and N4042 P-2. Specimen numbers: Plate 1, A–H: 398665–398672; Plate 2, A–X: 398673–398696.

**Reporting summary**. Further information on research design is available in the Nature Research Reporting Summary linked to this article.

## Data availability

The paleomagnetic and rock magnetic datasets generated and/or analyzed during the current study are available on the MagIC database repository site (https://earthref.org/MagicIC) and on Paleomagnetism.org 2.0 (https://api.paleomagnetism.org). Palynological slides are curated in the Museum of Paleontology, University of California, Berkeley, California, under the locality Nooitgedacht 68 – UCMP PA1378, with PA1378.01 (40.42 m) and PA1378.02 (29.9 m). Slides are N2990 P-1 and N2990 P-2, and N4042 P-1 and N4042 P-2. Specimen numbers: Plate 1, A–H: 398665–398672; Plate 2, A–X: 398673–398696. The authors declare that all additional data supporting the findings of this study are included in this published article (and its Supplementary Information, Figures, and Tables).

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

## Acknowledgements

Our research efforts were supported, in part, by: the Council for Geoscience (South Africa) to JN; JWG startup funds at the University of Texas–Dallas; and NSF EAR 0749895, 0934077, 1123570, and 1624302 to RAG. Field assistance by R. Jia and T. Stonesifer, Colby College, and S. Makubalo, Council for Geosciences, are acknowledged and appreciated. Anthony Hocking and the staff at the Royal Hotel Bethulie, South Africa, and A.J. Griesel, land owner, are acknowledged for their support of our efforts. Assistance in the paleomagnetism laboratory was provided by Z. Haque. Student participation was supported by the Selover Family student-research endowment and Barrett T. Dixon Geology Research and Internship Fund for undergraduate experiences in the Department of Geology, Colby College.

## Author contributions

R.A.G. and J.N. are responsible for measurement and description of the stratigraphic section, initial collection of the ash-fall bed for pilot analyses (2017), and collection of samples for palynological and Hg analyses. R.A.G. wrote all draft and revised manuscript versions which includes original and revised text supplied by co-authors. S.L.K. recollected the ash-fall bed (2018) and is responsible for U-Pb CA ID-TIMS analyses and interpretations, and all text associated with geochronology. J.W.G. collected mudstone and ash-fall bed samples for remanent magnetism, is responsible for all analyses and interpretations. He also collected the sample for petrographic inspection and conducted the petrographic analysis, and all text associated with rock magnetics. C.V.L. identified the elements of the palynological assemblages, is responsible for their interpretation, and all text associated with biostratigraphy and systematics. A.M.M. was responsible for analysis and interpretation of TOC and Hg data obtained from mudrocks collected across the purported PTB, as identified by Botha-Brink et al.[28], and the interval encompassing the air-fall ash bed, and all text on associated with Hg results. All co-authors read and contributed to all manuscript revisions.

## Competing interests

The authors declare no competing interests.
