## [Peer Review File · Nature Communications]

Reviewers' Comments:

Reviewer #1:

Remarks to the Author:

The terrestrial Permian-Triassic boundary sequences in the Karoo basin of South Africa has long been regarded as the references to explain the end-Permian mass extinction on land. It has been well exhibited by the turnover from the Daptocephalus to the Lystrosaurus Assemblages. However, it also has been seriously controversial whether this vertebrate turnover is coincident with the marine extinction precisely calibrated at the Meishan section in South China or not. Compared with the marine Meishan section, the terrestrial sections in the Karoo basin contain a much thick boundary interval from the Palingkloof Mb throughout the Katberg Fm with an uncertainty more than 40 m. Thus, it is hard to realize the correlation of the end-Permian mass extinction between terrestrial and marine ecosystems.

This paper for the first time provides a high-quality CA-ID-TIMS date from the lowest part of the Katberg Mb. The result confirms that the Triassic-type vertebrate Lystrosaurus began to occur in the latest Permian and some previously documented Permian-Triassic boundary in the Karoo basin was too old to be in the Permian. The first appearance of Lystrosaurus in the latest Permian is also consistent with the first appearance of some Triassic bivalves in the latest Permian. This high-precision date is extremely important to clarify the temporal relationship between the marine and terrestrial EPMEs. In addition, the normal polarity identified around the dated ash bed is also important for the correlation between the terrestrial and marine extinctions. The paper has the following weaknesses:

- 1) The high-precision TIMS date is extremely important, however, the date is clearly much older than the marine extinction interval between Bed 25 and 28. It is still not completely solved whether the terrestrial EPME in the Karoo basin is consistent with that at the marine section in South China.
- 2) Magnetostratigraphy is also very important, however, it seems the authors didn't sample the thickness enough, thus it is limited to be used for a wider correlation.
- 3) Like the magnetostratigraphy, the samples for mercury analyses are also probably insufficient. This is because the mercury anomalies previously reported from the PTB interval are all much younger than the dated ash bed in this paper (basically in the marine extinction interval). Thus, it is not surprised that the authors didn't see any anomaly of mercury at the section because the sampled interval is still probably in the background interval.

Overall, the paper contains both important geochronological and magnetostratigraphical data, both are worth to be published.

Reviewer #2:

Remarks to the Author:

The Karoo Basin hosts continental deposits with rich vertebrate faunal assemblages that span the Permian-Triassic boundary (c. 251 million years ago). These assemblages provide the world standard for understanding the patterns of faunal turnover in terrestrial ecosystems during the end-Permian extinction event (EPE) – Earth's deepest biotic crisis in the past 500 million years. Although a great deal of work has been carried out in the Karoo on vertebrate taxonomy and biostratigraphy, there has remained considerable controversy concerning the placement of the Permian-Triassic boundary in this basin and whether this is correlative with the mass extinction event, since absolute dating of the Karoo Basin succession has been scant. This new study, based on outcrop sections at Nooitgedacht, provides a detailed transect through the turnover between the Daptocephalus and Lystrosaurus zones, which marks the apparent major faunal turnover associated with the end-Permian crisis. Moreover, the new study provides welcome new high-precision radiometric data that constrains the age of the zone boundary to a minimum of 252+/-0.11 Ma. Additional studies of the palynology, palaeomagnetic signature and sediment geochemistry provides important data correlating the Karoo succession to

international reference sections. This study provides the first solid evidence that the terrestrial faunal turnover occurred several hundred thousand years before the official placement of the Permian-Triassic boundary, and also that the terrestrial faunal extinction occurred prior to the main pulse of marine faunal turnovers. Importantly, this study also supports recent work elsewhere in the Southern Hemisphere indicating that the terrestrial floral turnover also occurred well before the marine extinctions, and that the terrestrial floras and vertebrate faunas underwent more-or-less simultaneous turnovers in high southern latitudes. I commend the authors for their detailed, multidisciplinary approach.

I have only minor suggestions for improvement of the manuscript or comments on a few items that would be worth clarification:

1. Perhaps the title should read "The BASE OF THE Lystrosaurus Assemblage Zone, Karoo Basin, predates the" – since it appears that not the entirety of the Lystrosaurus Zone is within the Permian.
2. There is some ambiguity between the palynological samples mentioned in the text and those indicated on the logs (Fig. 3C and Fig. S1A,B). In the text, 2 productive samples are mentioned and one is indicated on Fig. 3C, four productive samples are mentioned in the supplementary data and indicated on Fig S1A,B but only two are described in the text. Did the other two productive samples contain only minimal diversity?
3. All the palynological samples come from just above the apparent end-Permian extinction level and are reasonably well correlated to the eastern Australian standards. Are there any palynological data from immediately below the faunal zone boundary (either from this or previous studies) to tie the pre-extinction beds to the Australian biozones?
4. Two additional papers, currently in press, may assist the palynostratigraphic correlations and palaeoenvironmental interpretations of the immediate post-extinction succession. These are: Vajda, V., McLoughlin, S., Mays, C., Frank, T., Fielding, C.R., Tevyaw, A., Lehsten, V., Bocking, M., Nicoll, R.S. (2020). End-Permian (252 Mya) deforestation, wildfires and flooding—An ancient biotic crisis with lessons for the present. *Earth and Planetary Science Letters* 529, xxx–xxx. <https://doi.org/10.1016/j.epsl.2019.115875>
Mays, C., Vajda, V., Fielding, C., Frank, T., Tevyaw, A., & McLoughlin, S. (in press). Refined Permian-Triassic floristic timeline reveals early collapse and delayed recovery of south polar terrestrial ecosystems. *GSA Bulletin* XX, xxx–xxx. DOI:10.1130/B35355.1
Of particular note, both of these papers note a marked pulse of algal remains in the immediate aftermath of the extinction event that reflects a degree of ponding in the landscape. The new palynofloras from the Karoo Basin are similarly rich in algae and are consistent with that interpretation.
5. The authors detected no marked Hg enrichment at the EPE in the Karoo Basin. It may be worth noting that the majority of previous studies identifying mercury spikes at the EPE are from Northern Hemisphere/Tethyan localities that would have been much closer to the putative source of Hg enrichment (Siberian trap magmatism/thermal metamorphism of organic matter). See, e.g.: J. Shen, T.J. Algeo, N.J. Planavsky, et al., 2019. Mercury enrichments provide evidence of Early Triassic volcanism following the end-Permian mass extinction. *Earth-Science Reviews*, <https://doi.org/10.1016/j.earscirev.2019.05.010>
Jun Shen, Jiubin Chen, Thomas J. Algeo, Shengliu Yuan, Qinglai Feng, Jianxin Yu, Lian Zhou, Brennan O'Connell & Noah J. Planavsky. 2019. Evidence for a prolonged Permian–Triassic extinction interval from global marine mercury records. *Nat. Comms.* <https://doi.org/10.1038/s41467-019-09620-0>
It is potentially a significant discovery that high southern localities were not significantly Hg-enriched.
6. Are there any identifiable plant macrofossils preserved in the studied succession that could be used to clarify whether the faunal and faunal turnovers were synchronous?
7. I wonder if the mineral grains labelled "augite" on Fig S2C might alternatively be green hornblendes. Augite would tend to indicate a fairly mafic composition of the ash bed, whereas hornblende could be present in ashes of felsic-intermediate composition, which are more common.

8. I have added a few comments on identifications to the pollen illustrations for the authors to consider.

9. I attach two pdfs with additional minor comments and corrections on the manuscript. The authors should check for consistency in the formatting of references.

Reviewer #3:

Remarks to the Author:

The manuscript "The Lystrosaurus Assemblage Zone, Karoo Basin, predates the end-Permian marine extinction: age and paleomagnetic evidence", by Gastaldo et al., presents a detailed biostratigraphic, paleomagnetic, and geochronological study of a key archive of terrestrial paleoclimate and tetrapod paleobiology across the Permo-Triassic boundary interval. The dramatic transitions in Earth Systems across this boundary, including mass extinction, extreme climate change, and large igneous province magmatism are of broad interest. The geophysical and geochronological methods are robust, and the authors are clearly experts in their fields of study, who have brought together an important multi-disciplinary data set and interpretation.

This is a very good manuscript of broad interest to those studying Earth systems transitions across the Permo-Triassic boundary. The authors' conclusions about the asynchrony of major terrestrial versus marine diversity fluctuations are firmly supported by the major new data reported, namely the U-Pb geochronology within a robust paleomagnetic and stratigraphic framework. However, while the data (and metadata) of this paper are carefully documented, I found lacking a key informative diagram which contrasts the relative timing of the Karoo basin events documented in this paper and their correlative events in Australia (palynology and floral change), south China (marine extinction), Siberia (flood basalt and intrusions). This seems to me to be a vital necessity for the broad readership of Nature Communications. The current Figure 2 could probably be relegated to the supplementary materials if room was needed.

My remaining comments on the manuscript appear rather minor in scope, but are important to how the content and conclusions of the manuscript are delivered.

a) Line 39: change "An" to "A"

b) Line 47: missing space between Lystrosaurus and AZ

c) Lines 53-54 and 57-58: There is some confusing repetition here about the interpretation that the study site preserves a "complete (terrestrial) Permo-Triassic Boundary (PTB) sequence". This stems perhaps from comparing earlier published claims to current interpretations, but it should be edited to be clearer and more concise.

d) Lines 56 and 60: There's a similar repetition in noting where this study picked the vertebrate biozone boundary?

e) Line 69: replace "such purported" with "these"

f) Line 80: This paragraph is muddled and partially incorrect... suggest to simplify as "Reported U-Pb dates are from the ^{238}U - ^{206}Pb decay scheme. This is the most robust system for geologically young rocks due to the greater abundance of ^{238}U and ingrown radiogenic ^{206}Pb . A summary of the U-Pb zircon isotopic data is presented in Table S1, and a concordia diagram and plot of individual $^{206}\text{Pb}/^{238}\text{U}$ zircon-crystal dates are presented in Figure 4."

g) Lins 96-100: The three data points collected using the ROM spike solution should be removed from the paper, as they are not clearly traceable to the higher quality data obtained using the EARTHTIME spike. This would require substantial documentation of spike intercalibration which is beyond the scope of this paper. This paragraph will need editing, but as pointed out by the authors on line 100 this will not substantially change the results and interpretation of the geochronological study.

h) Lines 101-104: remove the discussion of Th/U ratios and picking of analyses as this is a weakly

supported argument also has no effect on the results.

i) Line 136: the last sentence of this paragraph seems better suited to moving to line 134, in front of the sentence starting, "In contrast, elevated Hg:TOC values...".

j) Line 139: change "are" to "were"

k) Line 174: The paragraph structures of the discussion section are awkward; for example I would place the last sentence of the second paragraph, beginning with "Yet, our new high precision age determination...", as the first sentence of the subsequent paragraph, which is actually about the Australian comparison.

l) Line 190: Similarly I would truncate this very long paragraph, and start a new conclusion paragraph with the sentence, "The age obtained from the ash deposit..."

m) Line 179: delete "slightly older than, but"

The authors are to be congratulated on this excellent study; it makes the case that careful cm-by-cm stratigraphy in key archives of Earth system change can yield dramatic discoveries, and demonstrates the immense progress the field of integrated chronostratigraphy has made over the past decade.

My identity may be revealed to the authors.

Yours sincerely,

Mark D. Schmitz, Ph.D.

Professor, Department of Geosciences

We will address individual reviewer's concerns pertaining to our manuscript (NCOMMS-19-32552), in the following paragraphs. Several of the reviewers suggested changes in wording to the text which we have included in the revised version. Those changes appear as highlights in yellow; changes in the text that we have made as part of the current version appear in either orange or gray (conversion to Word from WordPerfect) highlight. To facilitate their review, I have included all text sent to us along with our responses.

Reviewer #1 (Remarks to the Author):

The terrestrial Permian-Triassic boundary sequences in the Karoo basin of South Africa has long been regarded as the references to explain the end-Permian mass extinction on land. It has been well exhibited by the turnover from the *Daptocephalus* to the *Lystrosaurus* Assemblages. However, it also has been seriously controversial whether this vertebrate turnover is coincident with the marine extinction precisely calibrated at the Meishan section in South China or not. Compared with the marine Meishan section, the terrestrial sections in the Karoo basin contain a much thick boundary interval from the Palingkloof Mb throughout the Katberg Fm with an uncertainty more than 40 m. Thus, it is hard to realize the correlation of the end-Permian mass extinction between terrestrial and marine ecosystems.

This paper for the first time provides a high-quality CA-ID-TIMS date from the lowest part of the Katberg Mb. The result confirms that the Triassic-type vertebrate *Lystrosaurus* began to occur in the latest Permian and some previously documented Permian-Triassic boundary in the Karoo basin was too old to be in the Permian. The first appearance of *Lystrosaurus* in the latest Permian is also consistent with the first appearance of some Triassic bivalves in the latest Permian. This high-precision date is extremely important to clarify the temporal relationship between the marine and terrestrial EPMEs. In addition, the normal polarity identified around the dated ash bed is also important for the correlation between the terrestrial and marine extinctions. The paper has the following weaknesses:

- 1) The high-precision TIMS date is extremely important, however, the date is clearly much older than the marine extinction interval between Bed 25 and 28. It is still not completely solved whether the terrestrial EPME in the Karoo basin is consistent with that at the marine section in South China.
- Shen et al. (2019) published a paper in GSA Bulletin about their South China section in which they report both marine and terrestrial environments. The paper was published contemporaneously with our submission. These authors document a succession dominated by marine deposits in which coastal deposits occur in the Penglaitan section in which fossil plants are preserved. One plant bed occurs in a black shale interpreted as having been deposited a ramp deposit which indicates that the assemblage is allochthonous. A second, low diversity plant-fossil horizon (N=1 taxon) occurs higher in a ramp-to-slope setting; this plant fossil also is allochthonous. These authors provide a U-Pb zircon age of 252.359 ± 0.038 Ma in the Penglaitan section D some 25+ m above the transported plant-fossil assemblage in section C. We are in full agreement with the reviewer that we are unable to solve the relationship between the Karoo record and that of the marine record in South China because it is not possible to infer terrestrial ecosystem dynamics from transported assemblages.
- 2) Magnetostratigraphy is also very important, however, it seems the authors didn't sample the thickness enough, thus it is limited to be used for a wider correlation.
- We presented a short magnetostratigraphic interval from Nooitgedacht in our previous draft because of several reasons. Our intent was to understand the polarity of the rocks in which the age date originated and determine how both related to the global latest Permian record. If the age estimate was consistent with the magnetozones of both Szurlies (2013) and Hounslow & Balabanov (2016), we could be confident of their correlation. If, though, our data indicated that the date fell in a reverse polarity zone, then we would have to explain this discrepancy. As we submitted the current manuscript, our PALAIOS contribution appeared (Gastaldo et al., 2019, 34:542– 561) in which we demonstrate that the turnover in vertebrate zones is found in a reverse polarity chron and not in a normal polarity chron. We conclude there that if a turnover in vertebrate assemblages occurred in the Karoo, that change was not coincident with the marine record which occurs in a normal polarity chron. Second, with the knowledge that our age assignment is within a normal polarity chron, it is possible to correlate it with the Bethel/Heldemoed/Donald 207 record where we have a robust magnetostratigraphy as recently documented by us, and correlate it more widely when both U-Pb age estimates are reported in a magnetostratigraphic context.
- We used a different technique in our sampling, employing oriented ceramic cubes rather than drill cores, to minimize the impact on the section. We were aware that other researchers had collected (Botha-Brink et al., 2014), and continued to collect (Botha et al., 2020) vertebrate material from the farm, and wanted to maintain confidentiality of our efforts. We now have

expanded our data set across a thicker interval surrounding the ash bed, from 5 m below the deposit to the base of the overlying sandstone. These data are presented in both the main text with additional text and figures residing in the supplemental information.

3) Like the magnetostratigraphy, the samples for mercury analyses are also probably insufficient. This is because the mercury anomalies previously reported from the PTB interval are all much younger than the dated ash bed in this paper (basically in the marine extinction interval). Thus, it is not surprising that the authors didn't see any anomaly of mercury at the section because the sampled interval is still probably in the background interval. Overall, the paper contains both important geochronological and magnetostratigraphical data, both are worth to be published.

- True. We now know that the age of the base of the *Lystrosaurus* AZ is late Permian and not early Triassic. At the time we sampled for Hg, we were testing the hypothesis of Botha-Brink et al. (2014) that the end-Permian event, coincident with that in the oceans, was represented by the bedded interval in their section. And, if, as claimed, the vertebrate-defined Permian–Triassic boundary was contemporaneous with the marine record, there should be a Hg anomaly in these rocks.

Reviewer #2

The Karoo Basin hosts continental deposits with rich vertebrate faunal assemblages that span the Permian-Triassic boundary (c. 251 million years ago). These assemblages provide the world standard for understanding the patterns of faunal turnover in terrestrial ecosystems during the end-Permian extinction event (EPE) – Earth's deepest biotic crisis in the past 500 million years. Although a great deal of work has been carried out in the Karoo on vertebrate taxonomy and biostratigraphy, there has remained considerable controversy concerning the placement of the Permian-Triassic boundary in this basin and whether this is correlative with the mass extinction event, since absolute dating of the Karoo Basin succession has been scant. This new study, based on outcrop sections at Nooitgedacht, provides a detailed transect through the turnover between the *Daptocephalus* and *Lystrosaurus* zones, which marks the apparent major faunal turnover associated with the end-Permian crisis.

Moreover, the new study provides welcome new high-precision radiometric data that constrains the age of the zone boundary to a minimum of 252±0.11 Ma. Additional studies of the palynology, palaeomagnetic signature and sediment geochemistry provides important data correlating the Karoo succession to international reference sections. This study provides the first solid evidence that the terrestrial faunal turnover occurred several hundred thousand years before the official placement of the Permian-Triassic boundary, and also that the terrestrial faunal extinction occurred prior to the main pulse of marine faunal turnovers. Importantly, this study also supports recent work elsewhere in the Southern Hemisphere indicating that the terrestrial floral turnover also occurred well before the marine extinctions, and that the terrestrial floras and vertebrate faunas underwent more-or-less simultaneous turnovers in high southern latitudes. I commend the authors for their detailed, multidisciplinary approach.

I have only minor suggestions for improvement of the manuscript or comments on a few items that would be worth clarification:

1. Perhaps the title should read “The BASE OF THE *Lystrosaurus* Assemblage Zone, Karoo Basin, predates the” – since it appears that not the entirety of the *Lystrosaurus* Zone is within the Permian.
 - Done. We agree that we have no sense of the upper limit of the AZ as currently defined, and we have eliminated two words following the colon because (1) punctuation is not allowed in the title and (2) the addition of “base of the” now exceeds the 15 word maximum title length.
2. There is some ambiguity between the palynological samples mentioned in the text and those indicated on the logs (Fig. 3C and Fig. S1A,B). In the text, 2 productive samples are mentioned and one is indicated on Fig. 3C, four productive samples are mentioned in the supplementary data and indicated on Fig S1A,B but only two are described in the text. Did the other two productive samples contain only minimal diversity?
 - As can be seen from our images, the quality of palynomorph preservation is not the greatest. We present data and images from the two horizons where we found the best preserved palynomorphs and greatest concentration from which we could assess diversity. The other two samples yielded low numbers of very poorly preserved pollen and spores.
3. All the palynological samples come from just above the apparent end-Permian extinction level and are reasonably well correlated to the eastern Australian standards. Are there any palynological data from immediately below the faunal zone boundary (either from this or previous studies) to tie the pre-extinction beds to the Australian biozones?
 - We wish there were intervals from which we recovered palynological assemblages at Nooitgedacht at one or more lower stratigraphic intervals. Although these may exist, we have yet to encounter them.
4. Two additional papers, currently in press, may assist the palynostratigraphic correlations and palaeoenvironmental interpretations of the immediate post-extinction succession. These are:

Vajda, V., McLoughlin, S., Mays, C., Frank, T., Fielding, C.R., Tevyaw, A., Lehsten, V., Bocking, M., Nicoll, R.S. (2020). End-Permian (252 Mya) deforestation, wildfires and flooding—An ancient biotic crisis with lessons for the present. *Earth and Planetary Science Letters* 529, xxx–xxx. <https://doi.org/10.1016/j.epsl.2019.115875>

Mays, C., Vajda, V., Fielding, C., Frank, T., Tevyaw, A., & McLoughlin, S. (in press). Refined Permian-Triassic floristic timeline reveals early collapse and delayed recovery of south polar terrestrial ecosystems. *GSA Bulletin* XX, xxx–xxx. DOI:10.1130/B35355.1
Of particular note, both of these papers note a marked pulse of algal remains in the immediate aftermath of the extinction event that reflects a degree of ponding in the landscape. The new palynofloras from the Karoo Basin are similarly rich in algae and are consistent with that interpretation.

- We have consulted both of these papers and incorporated their results in our text and supplemental information. We reached out to both Vivi Vajda (Vajda et al., 2020, *Earth and Planetary Science Letters* 529:115875) and Chris Mays (Mays et al., 2019, *GSA Bull.* 10.1130/B35355.1) and requested some clarification of the revised Australian palynozones they are now employing. These differ from those long held in Eastern Australia and to which we have correlated our palynological assemblages in the past (e.g., Gastaldo et al., 2015, 2017, 2019). We heard from Chris Mays on 5 January and Vivi Vajda on 19 January. In response to a few questions posed by Cindy Looy to each of them, we received definitive answers to some. In others, these authors are continuing to refine their biostratigraphy and are unable at this time (e.g., To directly answer your question: “In your Sydney Basin sections, does *Striatopodocarpites* disappear and *Protohaploxipinus*’ diversity decline starts at the base of your *P. crenulata* Zone, at the transition of the *P. crenulata* to *P. microcorpus* Zone, or more gradually in the *P. crenulata* Zone?”... I’m afraid that I have to postpone a solid answer at this point. We have incorporated the Eastern Australian palynozones in the revised manuscript, and have included this correlation on a new Figure 4 (as requested by Reviewer #3). We have noted the presence of an increased proportion of algal remains in the current version, consistent with what is reported from the Sydney Basin.

5. The authors detected no marked Hg enrichment at the EPE in the Karoo Basin. It may be worth noting that the majority of previous studies identifying mercury spikes at the EPE are from Northern Hemisphere/Tethyan localities that would have been much closer to the putative source of HG enrichment (Siberian trap magmatism/thermal metamorphism of organic matter). See, e.g.: J. Shen, T.J. Algeo, N.J. Planavsky, et al., 2019. Mercury enrichments provide evidence of Early Triassic volcanism following the end-Permian mass extinction. *Earth-Science Reviews*, <https://doi.org/10.1016/j.earscirev.2019.05.010>

Jun Shen, Jiubin Chen, Thomas J. Algeo, Shengliu Yuan, Qinglai Feng, Jianxin Yu, Lian Zhou, Brennan O’Connell & Noah J. Planavsky. 2019. Evidence for a prolonged Permian–Triassic extinction interval from global marine mercury records. *Nat. Comms.* <https://doi.org/10.1038/s41467-019-09620-0>

It is potentially a significant discovery that high southern localities were not significantly Hg-enriched.

- True. In our original supplemental document we state “There is the possibility that the Karoo Basin might be remote from a potential Siberian source area.” We have continued to include this probability in the supplemental document, and additional text and citations are now included. We also have unpublished Hg data from our 200 m section at Old Lootsberg Pass where we have an early Changhsingian age (Gastaldo et al., 2015, 2018) in which there are a number of very positive excursions from background levels indicating, most parsimoniously, contribution from Cape Fold Belt volcanism. We note, here, that Blewett and Phillips (2016: An overview of Cape Fold Belt geochronology: Implications for sediment provenance and the timing of orogenesis: *in* Linol, B., and de Wit, M., eds., *Origin and Evolution of the Cape Mountains and Karoo Basin: Regional Geology Reviews*, Springer Publishing, p. 49-55) identify a final pulse of volcanism in this tectonic belt in the Changhsingian based on ⁴⁰Ar/³⁹Ar results, from single muscovite grains and aggregate samples. It is unwise to attribute all volcanogenic deposits to the Siberian Traps without definitive and unique geochemical signatures originating from the Northern Hemisphere.

6. Are there any identifiable plant macrofossils preserved in the studied succession that could be used to clarify whether the faunal and faunal turnovers were synchronous?

- The occurrence of macrofloral elements is rare and highly dependent on the taphonomic factors influencing their preservation at this time (Gastaldo et al., 2005). We focused on the Loskop koppie because it is only here that we encountered the ash-fall horizon. We were unable to locate it on Spitskop and did not pursue sampling there. Botha et al. (2020) illustrate a low diversity *Glossopteris* assemblage, similar to what we’ve published elsewhere (e.g., Prevec et al., 2010; Gastaldo et al., 2017) low in the *Daptocephalus* AZ. They found no macrofloral evidence in the *Lystrosaurus* AZ on that hillside.

7. I wonder if the mineral grains labelled “augite” on Fig S2C might alternatively be green hornblendes. Augite would tend to indicate a fairly mafic composition of the ash bed, whereas hornblende could be present in ashes of felsic-intermediate composition, which are more common.

- John has returned to that thin section and, indeed, the mineral is not augite. Rather, it is amphibole, and this correction has been made in both the figure caption and supplemental text.

8. I have added a few comments on identifications to the pollen illustrations for the authors to consider.
- As noted above, Cindy was in email contact with both Vivi Vajda and Chris Mays about identifications and these have been considered.
9. I attach two pdfs with additional minor comments and corrections on the manuscript. The authors should check for consistency in the formatting of references.
- We have made the minor text modifications as suggested by the reviewer in this version, which are highlighted in yellow in the revised manuscript.

Reviewer #3

The manuscript “The *Lystrosaurus* Assemblage Zone, Karoo Basin, predates the end-Permian marine extinction: age and paleomagnetic evidence”, by Gastaldo et al., presents a detailed biostratigraphic, paleomagnetic, and geochronological study of a key archive of terrestrial paleoclimate and tetrapod paleobiology across the Permo-Triassic boundary interval. The dramatic transitions in Earth Systems across this boundary, including mass extinction, extreme climate change, and large igneous province magmatism are of broad interest. The geophysical and geochronological methods are robust, and the authors are clearly experts in their fields of study, who have brought together an important multi-disciplinary data set and interpretation.

This is a very good manuscript of broad interest to those studying Earth systems transitions across the Permo-Triassic boundary. The authors' conclusions about the asynchrony of major terrestrial versus marine diversity fluctuations are firmly supported by the major new data reported, namely the U-Pb geochronology within a robust paleomagnetic and stratigraphic framework. However, while the data (and metadata) of this paper are carefully documented, I found lacking a key informative diagram which contrasts the relative timing of the Karoo basin events documented in this paper and their correlative events in Australia (palynology and floral change), south China (marine extinction), Siberia (flood basalt and intrusions). This seems to me to be a vital necessity for the broad readership of Nature Communications. The current Figure 2 could probably be relegated to the supplementary materials if room was needed.

- We have developed a synthetic Figure 4, and moved Figure 2 to the supplemental information to accommodate it, in which we have made the requested correlations. We appreciate Mark's suggestion and feel that this summary diagram was a missed opportunity in our original submission.

My remaining comments on the manuscript appear rather minor in scope, but are important to how the content and conclusions of the manuscript are delivered.

- a) Line 39: change “An” to “A”
b) Line 47: missing space between *Lystrosaurus* and AZ

- Done. These changes are in yellow highlighted text.

c) Lines 53-54 and 57-58: There is some confusing repetition here about the interpretation that the study site preserves a “complete (terrestrial) Permo-Triassic Boundary (PTB) sequence”. This stems perhaps from comparing earlier published claims to current interpretations, but it should be edited to be clearer and more concise.

d) Lines 56 and 60: There's a similar repetition in noting where this study picked the vertebrate biozone boundary?

- We have added text indicating the presence of unconformities in the succession that indicate temporal gaps in the record, and changed the text for clarity. We note that since our submission, a new publication by Botha et al. (2020) has appeared in which the stratigraphic position of the PTB of these authors has changed from their original publication (Botha-Brink et al. 2014) ~8 m lower in the section. These authors provide no rationale for the lowering of the PTB, but now use this lower horizon (see revised Fig. 2 and Fig. S2 in which these interpretations are added) as the marine equivalent. Our U-Pb CA-ID-TIMS date now sits higher in this new scheme as proposed by Botha et al. (2020).

e) Line 69: replace “such purported” with “these”

- Done. These changes are in yellow highlighted text.

f) Line 80: This paragraph is muddled and partially incorrect... suggest to simplify as “Reported U-Pb dates are from the ^{238}U – ^{206}Pb decay scheme. This is the most robust system for geologically young rocks due to the greater abundance of ^{238}U and

ingrown radiogenic ^{206}Pb . A summary of the U-Pb zircon isotopic data is presented in Table S1, and a concordia diagram and plot of individual $^{206}\text{Pb}/^{238}\text{U}$ zircon-crystal dates are presented in Figure 4.”

- Change accepted and text wording adopted.

g) Lines 96-100: The three data points collected using the ROM spike solution should be removed from the paper, as they are not clearly traceable to the higher quality data obtained using the EARTHTIME spike. This would require substantial documentation of spike intercalibration which is beyond the scope of this paper. This paragraph will need editing, but as pointed out by the authors on line 100 this will not substantially change the results and interpretation of the geochronological study.

- We have decided to keep the 4 analyses determined with the ROM spike. Given the significance of our age result, the additional analyses only strengthen our interpretation that we have a magmatic population, and removing them has minimal (negligible) impact on the age interpretation. We also are not sure that “substantial documentation of intercalibration” is warranted when the data overlap with the Earthtime spike data.

h) Lines 101-104: remove the discussion of Th/U ratios and picking of analyses as this is a weakly supported argument also has no effect on the results.

- We have accepted Mark’s comment about the Th/U ratio lines and deleted them.

i) Line 136: the last sentence of this paragraph seems better suited to moving to line 134, in front of the sentence starting, “In contrast, elevated Hg:TOC values...”.

- Done.

j) Line 139: change “are” to “were”

- Change made.

k) Line 174: The paragraph structures of the discussion section are awkward; for example I would place the last sentence of the second paragraph, beginning with “Yet, our new high precision age determination...”, as the first sentence of the subsequent paragraph, which is actually about the Australian comparison.

- We’ve retained the sentence as the transitional (segue) sentence that bridges these two paragraphs.

l) Line 190: Similarly I would truncate this very long paragraph, and start a new conclusion paragraph with the sentence, “The age obtained from the ash deposit...”

- Done.

m) Line 179: delete “slightly older than, but”

- Done.

The authors are to be congratulated on this excellent study; it makes the case that careful cm-by-cm stratigraphy in key archives of Earth system change can yield dramatic discoveries, and demonstrates the immense progress the field of integrated chronostratigraphy has made over the past decade.

Figures

As noted above, we have moved our original Figure 2 to the Supplemental Information (Fig. S1) along with the attendant caption, and our stratigraphic column now appears as Figure 2. Modifications to this figure include the recent change in the horizon at which Botha and colleagues (Botha et al. 2020) now place their Permian–Triassic boundary, some 8 m lower in the stratigraphy than what was reported by them in 2014. This change in position, made without stating their rationale for the move, is noted in the diagram and caption. We have added the correlative palynological zones against our stratigraphy and note a gap between our highest assemblage in the *D. parvithola* zone and our lowest assemblage in the overlying *P. crenulata* zone. We also note that there is an erosional hiatus in the stratigraphy that is in the interval where we have been unable to recover palynomorphs.

We have constructed a figure in which we present a synthesis of what we know about the terrestrial record of megafloral, microfloral, and vertebrate remains in the Karoo Basin. This record is depicted in geochronological and magnetostratigraphic context

for the basin compared with (1) the palynological record, as currently understood, of Eastern Australia, (2) the timing and style of igneous activity associated with the Siberian Traps, (3) Late Permian magnetostratigraphy, and (4) the global chronostratigraphic time scale.

We hope that our revised manuscript is now acceptable for publication and look forward to a final decision about our contribution. On behalf of myself and my colleagues, we appreciate all your efforts in support of our continued research in one critical time of biodiversity loss in Earth history.

Respectfully,

Robert A. Gastaldo
Whipple-Coddington Professor of Geology

Reviewers' Comments:

Reviewer #3:

Remarks to the Author:

The content and editorial changes to the manuscript following peer review substantively address the suggestion and recommendations of the reviews, including my own. I support publication of this revised manuscript.